# Implementation of E-exams during the COVID-19 pandemic: A quantitative study in higher education

**Mohd. Elmagzoub Eltahir[1,2], Najeh Rajeh Alsalhi [1,2,3]\*, Sami Sulieman Al-Qatawneh[1,2]**

**1** College of Humanities and Sciences, Ajman University, Ajman, UAE, **2** Humanities and Social Sciences Research Center (HSSRC), Ajman University, Ajman, UAE, **3** Nonlinear Dynamics Research Center (NDRC), Ajman University, Ajman, UAE

\* n.alsalhi@ajman.ac.ae

**Data Availability Statement:** All relevant data are within the paper and its Supporting Information files.

**Funding:** This work was funded by a grant from the Deanship of Graduate Studies and Research at

## Abstract

The primary aim of this study was to identify the degree of acceptance of e-exams by undergraduate students at Ajman University during the spread of the COVID-19 pandemic. The study used the descriptive approach. A questionnaire consisting of 27 items was distributed to 1986 undergraduate students. The results of the study showed that undergraduate students demonstrated a moderate degree of acceptance of the implementation of e-exams during the spread of the COVID-19 pandemic, with females students finding them more acceptable than male students. Discipline and academic year also showed an impact on such acceptance, with Pharmacy & Health Science College students, and those in their third academic year demonstrating the highest levels of acceptance. Implications of the study raise awareness of the importance of addressing challenges associated with e-exams such as strict computer technology settings.

## 1. Introduction

The education system is witnessing great and successive developments to keep pace with the changes resulting from the advancement of science and technology and the implications of the spread of the COVID-19 pandemic. During the past few months, most national governments have shut down in-person teaching in their educational institutions completely or partially to curb the rapid spread of COVID-19 [1, 2]. Educational institutions and systems have therefore sought to adapt to these developments through teaching and evaluation strategies suitable to this new environment of the COVID-19 pandemic [3]. The evaluation process is among the main components of any stage of the educational process; because through it students are sorted according to their abilities and their achievement progress [4]. Indicates that improving and developing evaluation methods is one of the five goals of the academic community, as international academic accreditation institutions such as the North Central Association of Colleges and Schools (NCA) and the National Council for Accreditation of Teacher Education (NCATE) consider evaluation as one of the basic and necessary criteria for accreditation. E-exams came to provide a great service to the education system during the spread of a COVID-19 pandemic. E-exams also represent other benefits. Faculty members save time and effort,

Ajman University, Grant No. (2021-IRG-HBS-4). The funders had no role in study design, data collection and analysis, decision to publish, or preparation of the manuscript.

**Competing interests:** The authors have declared that no competing interests exist.

and students are safer with these e-exams when compared with traditional printed paper tests: the latter require a lot of time and effort to correct them and extract the results and announce them to students [5]. In addition, e-exams are considered one of the most important e-learning tools that measure achievement in developed countries [6]. Technology has enabled modern, unconventional evaluation methods, such as computerized evaluation, online assessment, remote evaluation, and question banks. Nowadays, during the COVID-19 pandemic, the e-exams system becomes looked to be a rapidly developing assessment instrument due to its precision and reliability [7]. According to [8], most educational institutions started using the e-exams system during the COVID-19 pandemic due to its positive features, such as reductions in the time required for students' exams and institutions easily monitoring students during their examination. However, students' perspectives on its implementation in Emirati universities such as Ajman University remain unexplored. Thus, the study sought to investigate the implementation of e-exams during the COVID-19 pandemic in higher education institutions in the UAE. E-exams were carried out at Ajman University in the UAE, in the fall of 2020 during the spread of COVID-19. The current study is therefore aimed at investigating students' acceptance of the implementation of e-exams in their university.

Prior to the COVID-19 pandemic, the issue of e-exams had already been addressed in a variety of previous studies [9–11] that aimed to explore and identify attitudes and perceptions of students and faculty members regarding e-exams or e-assessments are very important because they allow us to predict and interpret behavior in the future. Consequently, a decisive decision may have to be made regarding the system and mechanism for the assessment and evaluation of students, whether e-examinations will be used in lieu of traditional methods of assessment, or if they will be reduced or eliminated entirely. The authors hope in their current study on the impact of the transformation of most educational institutions in the countries of the world from face-to-face or traditional learning to blended learning, which represents a sudden shift to distance learning, which may lead to extending towards a more open digital ecosystem for e-exams. Moreover, E- exams have undergone significant progress, are now ubiquitous among higher education institutions around the world, and are rapidly being preferred due to the advent of the COVID-19 pandemic. Thus, this investigation will help officials in higher education institutions and universities to make appropriate decisions about the permanent adoption of e-exams during the COVID-19 pandemic and the possibility of their application after the pandemic. An additional aim is to give university officials feedback on the level of students' acceptance of e-exams during the COVID-19 pandemic. This means that the current study may supply higher education institutions with sufficient information about students' degree of acceptance of the e-exams implemented during the spread of the COVID-19 pandemic. In turn, this will assist in the adoption of e-exams as a reliable assessment instrument and a valid alternative to traditional printed examinations in higher education institutions in the future [3]. In order to explore Ajman University undergraduate students' degree of acceptance of e-exams during the spread of the COVID-19 pandemic, the study seeks to answers to the following research questions:

**RQ1:** To what extent do Ajman University undergraduate students accept e-exams during the spread of the COVID-19 pandemic?

**RQ2:** Does the degree of Ajman University undergraduate students' acceptance of e-exams during the spread of COVID-19 vary according to gender, college, and academic year?

## 2. Literature review

### 2.1 Definition of E-exams (E-exams)

E-exams can be defined as all forms of assessment and evaluation that are carried out using digital technologies [6, 12, 13]. It is a process that includes the implementation of exams

through the internet or intranet [14] (Elsalem et al., 2021) [1]., 2021defined e-exams as computer-based and internet-based, where questions are posed to students, corrected directly, with feedback provided on the student's responses and scores reported, and appropriate security measures taken to maintain confidentiality. Moreover [15], defined e-exams as a timed, controlled, summative evaluation exam carried out using each candidate's own device working a unified operating system. According to [16], e-exams have benefits compared to traditional paper exams, such as that they may include new multimedia and be interactive, and software test elements that have greater validity for it, in addition to ease of labeling, being time-saving, minimizing managerial expenses, and achieving reductions in the cost of raw materials.

## 2.2 E- exams requirements, criteria & training the faculty members

According to [17], there are five requirements overlapping for an e-exam, which are; Students preparing for the exam, topics of the exam, a software package, electronic display sets, and a fully equipped facility. Fig 1. illustrated these requirements.

Moreover [18], referred to significant processes when implementing e-assessment in higher education: to determine intended recipients and the purpose of testing; select appropriate instruments and e- platform; specify forms of feedback; clarify the tasks; provide a knowledge base for operating e-tests, etc. From the other side [19], pointed out that there are eight criteria for developing and implementing electronic tests, including development and implementation of design options, scalability, security, accessibility, and usability. and features of feedback, uniqueness, cost, and interoperability. Furthermore [20], describe five essential criteria of e-exams which are as the following:

- *Strategic*: this means that considers the identification of the key elements for improvement based on the acquisition of competencies.

- *Integral*: this means that assures the integral acquisition of the competencies.

- *Holistic*: this means that considers all the internal and external agents.

- *Transversal*: this means that affects all of the learning actions and activities and the interactions that take place during the learning process.

- *Coherent*: this means that considers the different processes as interrelated and not isolated, giving coherence to the assessment.

There is no question that the Learning Management Systems (LMS) programs (software) of e-learning in educational institutions and training learners on it is a motivating element for both the teacher and the learner to make use of the Internet in the educational process. These systems are designed to help teachers use the Internet in teaching, assessment and communication with students in an easy way without the need for deep knowledge of programming

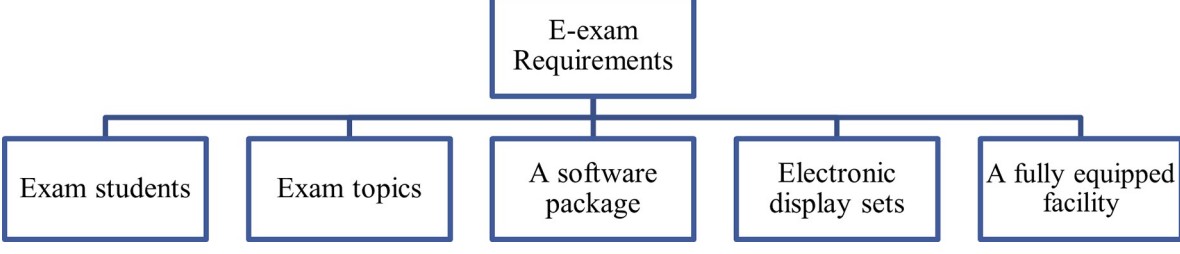

**Fig 1. E-exam requirements.**

methods. Examples of such LMS: Blackboard, Schoology, Dokeos, ATutor, Moodle, Web CT, Leapsome, Canvas, Brightspace, etc.

Ajman University in the Emirates adopted Moodle as one of the Learning Management Systems (LMS) programs during the COVID-19 pandemic. Where the University adopted online learning, and students were evaluated through e-exams, especially after heavily relying on e-exams during the Covid-19 pandemic. Therefore, the university held training and workshops for the faculty members to provide them with basic skills and proper knowledge about designing and preparing for e-exams. Fig 2 illustrates the use of Moodle by faculty members of all colleges at Ajman University of their designing and preparation the e-exams during the COVID-19 pandemic during the second semester of the 2019/ 2020 academic year.

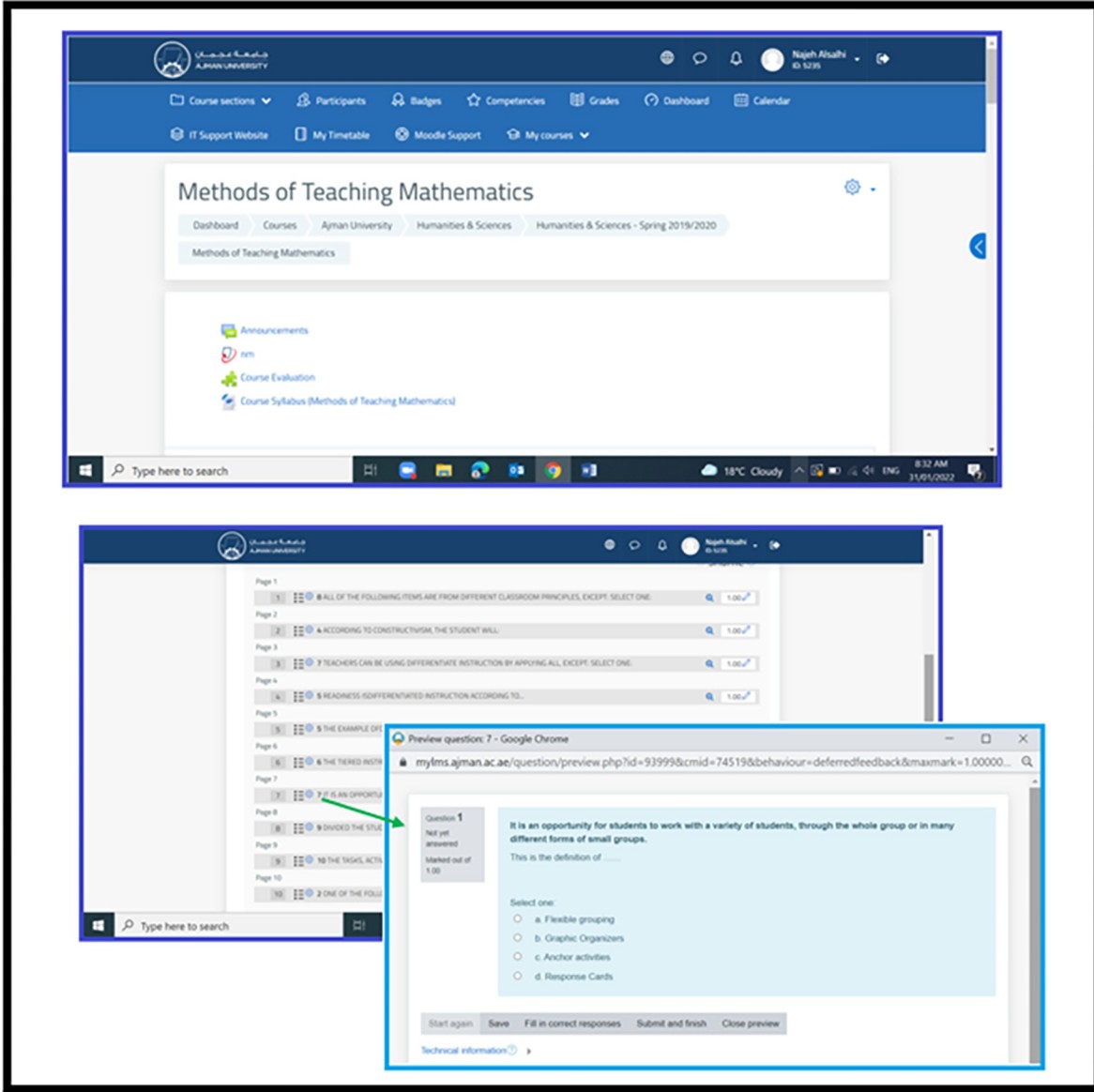

**Fig 2. Using Moodle in preparing and designing e-exams.**

## 2.3 E- exams benefits

There is a group of studies that highlight a number of the main benefits of online e-exams, in addition to some noticeable challenges from the point of view of both teachers and students, when compared to traditional printed paper exams [4, 21–24]. These can be summarized as seen in Fig 3.

Several studies pointed out that students prefer e-exams over traditional printed exams because they can take the exam at any time and anywhere, they can obtain feedback and marks more quickly, they have more control, are easy and quick to use, save time, and are environmentally friendly [25–27]. In contrast, we find that traditional paper exams have numerous disadvantages, such as being time-consuming. Grading of paper-based exams needs to be done manually, which is difficult and laborious. The examination of many students requires more invigilators. Grades are not exact, as calculations are carried out manually. Examination results can be lost, and it takes a long time to check the results, as this is done manually.

## 2.4 Challenges related to E-exams during the COVID-19 pandemic

After campus closures to curb the spread of COVID-19 pandemic, universities and schools have faced many challenges related to changing the system of learners' exams to e-exams instead of traditional paper exams. There are many challenges to the implementation of e-exams, according to the Organization for Economic Co-operation and Development [6, 8, 17, 28]. As shown in Fig 4, they can be summarized as follows:

## 2.5 Electronic exam usability

In reviewing the linked literature and studies, the authors found that research about the application of e-exams in universities and schools remains insufficient to justify their success and effectiveness in education and learning as a replacement for traditional printed exams. There have been several studies conducted by universities and academic institutions on the use of e-exams in their institutions, all of which centered on the features and the benefits of e-exams [1–3, 7, 8, 14, 18, 23, 24, 29–34]. Results from these studies focused on the perceptions of

**Benefits of E-exams**

- Constitute a fully integrated system that enhances the validity of evaluations using improved question forms that integrate interactivity and multimedia.
- Improve the reliability of the performance and the robu stness of the test results, enabling detailed analysis.
- Provide direct feedback, easily correcting misconceptions.
- May substantially enhance the accuracy of data management tasks like labeling, moderating, and storing the data.
- Decreasing the workload of teachers through saving time spent on routine work.
- Reducing significantly the burden of monitoring while examining huge student populations.
- Economical since it saves effort, time and money.
- Flexibility since it can be applied before, after or during the explanation.

**Fig 3. Benefits of e-exams.**

| Inexperienced undergraduate students with the E-exams | • Undergraduate students need a training at the beginning to be familiar with E-exams. |
|---|---|
| Possible student academic dishonesty | • In reality, this means and translates to cheating and plagiarism, which is by far the most widely debated problem in educational institutions today with regard to the change to electronic exams. |
| Assessing and certifying practical knowledge and skills | • In certain respects, the evaluations of learners cannot be based on written exams to evaluate the learned knowledge and skills, because it requires a direct evaluation of the real ability of learners to perform a specific task. |
| Ensuring fairness | • All the learners meet in the same room during on-site exams to take an exam under similar conditions (timing, material available, etc.). In contrast, learners do not benefit from the same working conditions when taking off-site online tests. |
| Risk of technical failure | • This means that good and efficient LMS servers and systems are needed to ensure students and faculty members have access to the Internet from their homes. |

**Fig 4. Challenges related to e-exams during COVID-19 pandemic.**

students and faculty members concerning the relative benefits, features, and challenges of e-exams and their efficiency compared to traditional paper exams. The results showed that students' showed their openness to and acceptance of the e-exams. They also confirmed that they prefer electronic computerized tests of the type of multiple-choice questions; in addition to a preference for the feature in e-exams that enables them to re-sit the exam several times in order to improve their scores. Additionally, in [34], the results indicated that the application of e-exams does not have a negative impact on students' grades and academic achievement, and features of e-exams were appropriate and accepted by students [35]. Reported that the system of electronic examinations could reduce the burden on teachers and enhance instructional quality. Also, studies have confirmed that e-exams offer direct feedback to students and help improve learning in comparison with traditional paper exams [26, 27, 36]. On the other hand, some other studies have shown that students were upset about the inability to explain their responses and answers because of strict computer technology settings, which raised their stress and confusion during the exam [37]. At the same time, some studies have shown that students' readiness for type of test the need to complete, together with the consistency of the exam, eventually affects their academic results [38]. According to [39], there are reasons for removing the time-limits imposed on e-exams during the COVID-19 pandemic: it causes unnecessary pressure on students; problems with bandwidth and network connectivity can cause delays; timed exams measure speed, which is only weakly connected with comprehension; is not secure from fraud; and it might put an extra burden on learners who really need learning accommodations. Moreover [9], pointed out that while e-exams might be a reliable tool to measure what they aim to measure, they may also raise learners' anxiety and tension levels and might makes it easier to cheat. On the other hand [14], pointed out that learners preferred print to digital displays during their studying and exams, as reading electronic online exams may lead to

more cognitive stress load on the reader compared to reading from print exams [40]. Also revealed that learners thought it easier to cheat on electronic examinations conducted online.

The current study differs from the previous studies in that it attempts to investigate the degree of perspective of undergraduate students to accept e-exams at Ajman University as one of the higher education institutions in all colleges in the university and its number (9), which means that it includes all disciplines of undergraduate students. Moreover, this study examined the degree of acceptance of undergraduate students at Ajman University for e-exams during the spread of COVID-19 according to gender, college, and academic year variables.

## 2.6 Significance of study

- Better understanding of students' perspectives on e-exams will assist in identifying the major challenges in achieving undergraduate students' acceptance of their application.

- The findings of the current study might encourage and facilitate the sustained switchover to E-learning and e-exams as an assessment process in the education system during and after COVID-19.

- Findings of the current study might benefit higher education institutions and other educational sectors outside of the United Arab Emirates.

## 3. Method

### 3.1 Approach of the study

The current analysis was carried utilizing a descriptive method approach, which is a type of research that describes an under-examined population, condition, or phenomena by gathering quantifiable data that can be used for statistical analysis [41]. According to [42], the prime purpose of descriptive research is to examine phenomena and their specific features. Moreover, it deals with what really occurred, instead of why or how Thus, a questionnaire instrument was utilized to gather data from a sample of the population.

### 3.2 Population of study

The research population comprised male and female students of all Ajman University colleges registered in the second semester in the academic year 2019/2020. The total number was 6620 undergraduate students as shown in Table 1 and Fig 3.

**Table 1. Study population.**

|  | College | # of students | (%) |
|---|---|---|---|
| 1 | College of Dentistry | 944 | 14.3% |
| 2 | College of Pharmacy & Health Sciences | 397 | 6.0% |
| 3 | College of Engineering and Information Technology | 1209 | 18.3% |
| 4 | College of Architecture, Art and Design | 531 | 8.0% |
| 5 | College of Business Administration | 979 | 14.8% |
| 6 | College of Law | 545 | 8.2% |
| 7 | College of Mass Communication | 589 | 8.9% |
| 8 | College of Humanities and Sciences | 1304 | 19.7% |
| 9 | College of Medicine | 122 | 1.8% |
| | **Total** | **6620** | **100.0%** |

**Table 2. Research sample.**

|  | College | # of students | Percentage (%) |
|---|---|---|---|
| 1 | Dentistry | 283 | 14.2% |
| 2 | Pharmacy & Health Sciences | 119 | 6.0% |
| 3 | Engineering and Information Technology | 363 | 18.3% |
| 4 | Architecture, Art and Design | 159 | 8.0% |
| 5 | Business Administration | 294 | 14.8% |
| 6 | Law | 163 | 8.2% |
| 7 | Mass Communication | 177 | 8.9% |
| 8 | Humanities and Sciences | 391 | 19.7% |
| 9 | Medicine | 37 | 1.9% |
| **Total** | | **1986** | **100.0%** |

## 3.3 Sample

A sample of 30% of the population of each college was taken by the investigators. A random sampling method, implemented through a stratified sample technique, was used to obtain the sample for this study, which totaled 1986 (6620 * 30/100 = 1986) students. For example, for the students of the College of Dentistry, 944 * 30/100 = 283.2, which indicated that a sample of 283 students was required from this college. As a percentage of the total sample, College of Dentistry students were 283 /1986* 100 = 14.2%. The same process was followed for the other colleges (see Table 2).

A total of 1986 electronic questionnaires were designed and distributed via emails, WhatsApp, and social media to students in order to collect the data needed to achieve the study objectives. Of these, 1742 were returned completed correctly and in full. A number of learners (n = 244) across all selected colleges did not responding correctly to the questionnaire. Consequently, the sample became 1742 students. Table 3 shows the demographic data for the selected sample of students who answered the questionnaire correctly.

**Table 3. Demographic information of students.**

| Study variables | Variables levels | Frequency (f) | Percentage (%) |
|---|---|---|---|
| **Gender** | Female | 964 | 55.3% |
|  | Male | 778 | 44.7% |
|  | **Total** | **1742** | **100.0%** |
| **College** | Dentistry | 253 | 14.5% |
|  | Pharmacy & Health Sciences | 112 | 6.4% |
|  | Engineering and Information Technology | 295 | 16.9% |
|  | Architecture, Art and Design | 146 | 8.4% |
|  | Business Administration | 264 | 15.2% |
|  | Law | 161 | 9.2% |
|  | Mass Communication | 144 | 8.3% |
|  | Humanities and Sciences | 334 | 19.2% |
|  | Medicine | 33 | 1.9% |
|  | **Total** | **1742** | **100.0%** |
| **Academic year** | First | 542 | 31.1% |
|  | Second | 447 | 25.7% |
|  | Third | 285 | 16.4% |
|  | Fourth | 234 | 13.4% |
|  | Fifth | 234 | 13.4% |
|  | **Total** | **1742** | **100.0%** |

### 3.4 Study instrument

The questionnaire was used to gather data from the sample learners. It was sent to them during the second semester of the academic year 2019/2020, during the occurrence of the COVID-19 pandemic. During the design of the questionnaire, similar research in this area was reviewed, such as studies conducted by [43]. The questionnaire comprised of two sections. The first section concerned students' general information, and the second part represented the questionnaire elements (n = 27) based on the study's objectives.

**3.4.1 The validity of the instrument.** A group of arbitrators (10) faculty members of UAE universities) with extensive experience in the field of education were asked to express their views on the items of the questionnaire, in terms of the relevance of items for achieving the research aims and the number and comprehensiveness of the questionnaire items. The educational specialists' comments and suggested modifications were taken into account, and relevant deletions, amendments, and additions were made. As a result, the questionnaire after modification consisted of 27 elements, to achieve the objective of the research.

**3.4.2 Reliability of the instrument.** To verify the internal consistency of the study tool, Cronbach's α was used. It was applied to a pilot study involving 50 students from outside the study sample, for which the calculated Cronbach alpha coefficient was 0.874.

### 3.5 Ethical considerations

This study was approved by the Research Ethics Committee/Deanship of Graduate Studies and Research of Ajman University (Reference number: H-H-F-2020-May-28) on May 30, 2020.We obtained informed consent was obtained from all individual participants included in the study.

### 3.6 Data analysis measures

In this analysis, a five-dimensional Likert scale is implemented, as shown in Table 4 below.

### 3.7 Statistical analysis of the data

For data analysis, the researchers utilized the Statistical Package for the Social Sciences (SPSS) to compute the percentage, mean, standard deviation SD, independent t-test tests, one-way ANOVA, and the Scheffe test.

## 4. Results

### 4.1 Findings of the study attributed to question 1: To what extent do Ajman University undergraduate students accept e-exams during the spread of the COVID-19 pandemic?

To address the first research question, we computed average scores and standard deviations of participants' responses to every one of the Items 1–27, which were relevant to the students' acceptance of e-exams during the spread COVID-19 pandemic, as seen in Table 5.

**Table 4. Evaluation of scale data based on the options of scale and score intervals.**

| Description | Scores | Intervals |
|---|---|---|
| Very high | 5 | 4.21–5.00 |
| High | 4 | 3.41–4.20 |
| Moderate | 3 | 2.61–3.40 |
| Low | 2 | 1.81–2.60 |
| Very low | 1 | 1.00–1.80 |

**Table 5. Descriptive statistics for the students' responses to the items about the degree of acceptance of E-exams.**

| No. | Paragraphs | Mean | SD | Description |
|---|---|---|---|---|
| I1 | E-exams provide a more engaging experience than using paper | 3.23 | 1.34 | Moderate |
| I2 | E-exams are more environmentally friendly than paper exams | 3.68 | 1.25 | High |
| I3 | E-exams provide the ability to easily identify and access unanswered questions | 3.62 | 1.09 | High |
| I4 | I think that e-exams are more familiar for me than printed paper exams | 3.45 | 1.11 | High |
| I5 | The number of electronic exam questions is sufficient | 3.56 | 1.11 | High |
| I6 | The electronic exam system is clear and specific | 3.59 | 1.11 | High |
| I7 | E-exams help extract results quickly, meaning feedback and marks are provided more quickly | 4.13 | 1.00 | High |
| I8 | Electronic exam regulations are clear and easy to understand | 3.64 | 1.09 | High |
| I9 | Electronic exam times are appropriate for students | 2.68 | 1.17 | Moderate |
| I10 | I think the electronic exam system was successful in protecting against technical problems | 2.54 | 1.16 | Moderate |
| I11 | Students do not need external help when using the computer [for e-exams] | 2.57 | 1.32 | Moderate |
| I12 | The exam time is not enough to answer all questions | 3.21 | 1.33 | Moderate |
| I13 | E-exams help raise the efficiency of student achievement | 3.01 | 1.33 | Moderate |
| I14 | E-exams serve as an accurate and reliable assessment method | 2.67 | 1.29 | Moderate |
| I15 | I would recommend the e-exams system to others | 2.91 | 1.34 | Moderate |
| I16 | E-exams are suitable for assessing students on any course | 2.90 | 1.31 | Moderate |
| I17 | Taking the electronic exam requires less time than taking the paper-based exam | 2.87 | 1.25 | Moderate |
| I18 | I prefer taking paper-based exams for assessing my knowledge | 2.96 | 1.23 | Moderate |
| I19 | E-exams enable me to show a better academic achievement | 3.04 | 1.36 | Moderate |
| I20 | E-exams serve as a flexible assessment method | 3.00 | 1.33 | Moderate |
| I21 | E-exams make me feel less stressed than paper-based exams | 3.29 | 1.27 | Moderate |
| I22 | I feel that the program (software) in the e-exams system is easy to use and deal with | 3.27 | 1.21 | Moderate |
| I23 | In general, I prefer taking e-exams to taking paper-based exams | 2.94 | 1.22 | Moderate |
| I24 | Internet interruption while I am doing e-exams causes me great anxiety | 3.72 | 0.99 | High |
| I25 | Electronic online exams make me feel more stress, pressure, and anxiety compared to printed paper-based exams | 3.15 | 1.30 | Moderate |
| I26 | I feel that it is easy to cheat while performing e-exams | 3.70 | 1.03 | High |
| I27 | I think that e-exams are more difficult than traditional exams | 2.56 | 1.32 | Moderate |
| | **Total** | **3.18** | **1.22** | Moderate |

The findings shown in Table 5 show that the mean for responses for all items (1–27) was 3.18 (SD 1.22), indicating that the students showed a moderate acceptance of e-exams during the COVID-19 pandemic. This finding might indicate that some students still prefer traditional paper exams, even though there is a COVID-19 pandemic. It is also evident from Table 5 that the students' answers to Item-7 ('E-exams help extract results quickly, meaning feedback and marks are provided more quickly') was given the highest mean value (4.13) at a high degree, and Item-24 ('Internet interruption while I am doing e-exams causes me great anxiety') came in second, also at a high level with a mean value of 3.72. Item-26 ('I feel that it is easy to cheat while performing e-exams') came in third, at a high level with a mean value of 3.70. Moreover, Item-2 ('E-exams are more environmentally friendly than paper exams') came in fourth, also at a high degree of acceptance of e-exams with a mean value of 3.68. Furthermore, it is also evident from the students' responses to Item-8 ('Electronic exam regulations are clear and easy to understand') that this question was rated as having the fifth highest degree of acceptance of using e-exams, with a mean of 3.64, and at a high degree. Similarly, a high degree was also found for Items 3, 4, 5, and 6 with the respective mean values of 3.62, 3.45, 3.56, and, 3.59. The lowest mean (2.54) was acquired for Item-10 ('I think the electronic exam system was successful in protecting against technical problems'), suggesting a moderate degree. In the same way, a moderate degree also obtained for Qs 1, 9, 11, 12, 13, 14, 15, 16, 17,

18, 19, 20, 21, 22, 23, 25, and 27, with the respective mean values of 3.23, 2.68, 2.57, 3.21, 3.01, 2.67, 2.91, 2.90, 2.87, 2.96, 3.04, 3.00, 3.29, 3.27, 2.94, 3.15 and 2.56.

### 4.2 Findings of the study attributed to question 2: Does the degree Ajman University undergraduate students' acceptance of e-exams during the spread of COVID-19 vary according to gender, college, and academic year?

Mean scores and SD were calculated for questions, and t-test, one-way ANOVA test, and Scheffe's post-hoc comparison test were also conducted to determine the significance of the variations between averages. The findings of the answers to the study subjects are listed below according to the study variables.

**4.2.1 First: Gender variations among students.** A t-test was utilized to assess the significance of the differences between the averages of the acceptance of e-exams by undergraduate students at Ajman University during the spread COVID-19, from the perspective of students, according to gender, as shown in Table 6.

As presented in Table 6 and Fig 5, the findings clearly illustrated that the computed t value was 2.037, which is greater than the (t) table, indicating the presence of significant differences between the mean values for males and females (in favor of females), at the significance level of 0.042, which is less than the required statistical significance level (0.05). The finding means that female Ajman University undergraduate students are more accepting than their male counterparts of e-exams during the spread of COVID-19.

**4.2.2 Second: College variable among students.** A one-way ANOVA test was utilized to assess the significance of the differences between averages of Ajman University undergraduate students' acceptance of e-exams during the spread COVID-19, according to college variable among students. The findings of the one-way ANOVA test of this variable are shown in Table 7 and Fig 7. As displayed in Table 7 and Fig 6, the results clearly illustrated that there are statistically significant differences in students' perspectives according to the variable of college, as the p-value is 0.003, which is less than the required statistical significance level (0.05).

Therefore, in order to identify the source of the differences, the Scheffe test was used for the following comparisons, and the findings are shown in Table 8 below. The results shown in Table 8 emphasize that the source of the differences in the students' acceptance of e-exams according to the variable of college was in favor of students of the Pharmacy & Health Science College.

**4.2.3 Third: Academic year variable.** A one-way ANOVA test was utilized to assess the significance of the differences between averages of the acceptance of e-exams by undergraduate students at Ajman University during the spread COVID-19, according to academic year variable. The findings of the one-way ANOVA test of this variable are shown in Table 9 and Fig 7. As displayed in Table 9 and Fig 7, the results clearly illustrate that there are statistically significant differences in students' perspectives according to the variable of academic year, as the p-value is 0.003, which is less than the required statistical significance level (0.05).

Therefore, in order to identify the source of the differences, the Scheffe test was used for the following comparisons, and the findings are shown in Table 10 below. The results shown in

**Table 6. Means and SD of the students' answers based on gender.**

| Gender | N | Mean | SD | Mean Difference | T. Value | df | Sig. |
|--------|---|------|-----|-----------------|----------|-----|------|
| Female | 42 | 3.06 | .503 | 0.05814 | 2.037 | 1740 | 0.042* |
| Male | 45 | 3.20 | .723 | | | | |

* Statistically significant at (p<0.05)

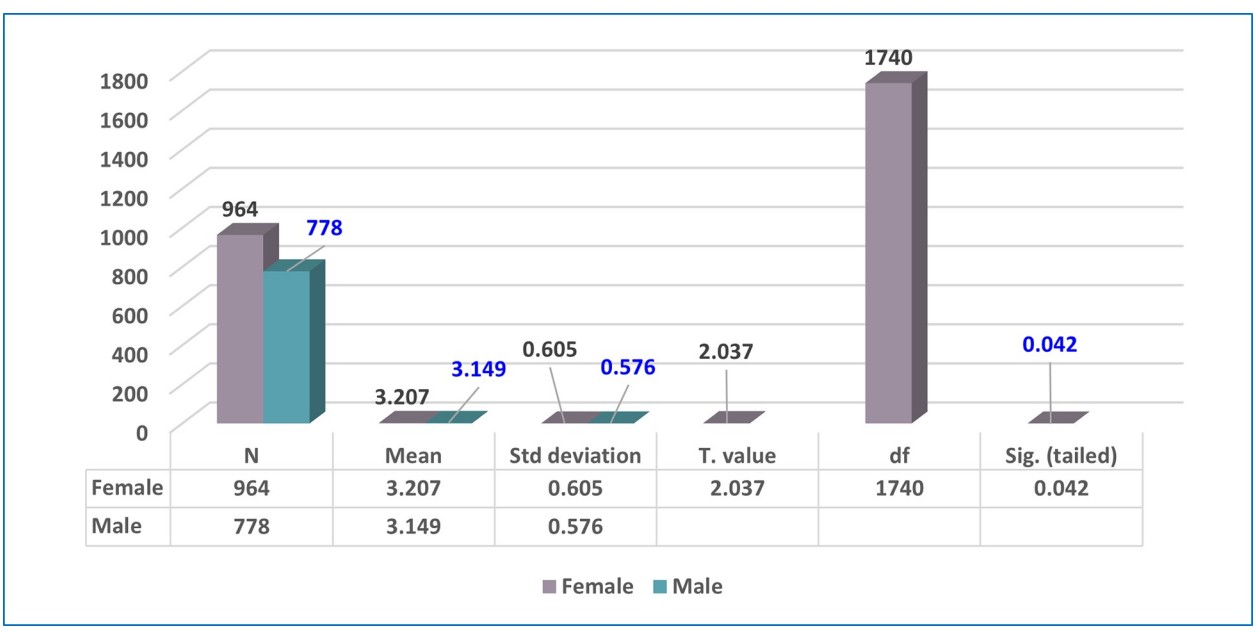

**Fig 5. Means and standard deviations of the students' answers based on gender.**

Table 10 indicate that the source of the differences in the students' acceptance of e-exams according to the variable of academic year was in favor of students in their third academic year.

## 5. Discussion

Results of the first research question on the degree of acceptance of e-exams during the spread of the COVID-19 pandemic at Ajman University indicate that, from the students' own perspective, the degree of acceptance of e-exams was at a moderate level, with a general arithmetic mean of 3.18 and standard deviation of 1.22. The moderate result might mean that some of the undergraduate students at Ajman University accepted the implementation of e-exams during the COVID-19 pandemic, while others did not, preferring the traditional paper-based exams. Based on the results in Table 5, related to the students' responses to the questionnaire items, it was noted that some of their responses indicated positive attitudes towards the implementation of e-exams at Ajman University during the Covid-19 pandemic. Items Item-2, Item -3, Item -4, Item -5, Item -6, Item -7 and Item -8 all indicated high degrees of acceptance. This means that undergraduate students might be satisfied to accept the implementation of e-exams in their university during the Covid-19 pandemic spread, which may be due to reasons related to the features of e-exams such as quicker feedback and marks, saving time, environmentally friendly, easy to identify and access unanswered questions, the system of e-exams

**Table 7. One-way ANOVA test for college variable among students.**

|  |  | Sum of squares | df | Mean square | F | Sig. (tailed) | Sig. level |
|---|---|---|---|---|---|---|---|
| College variable | **Between Groups** | 16.394 | 8 | 2.049 | 5.966 | 0.001 | Significant |
|  | **Within Groups** | 595.321 | 1733 | 0.344 |  |  |  |
|  | **Total** | 611.715 | 1741 |  |  |  |  |

* Statistically significant at (p<0.05)

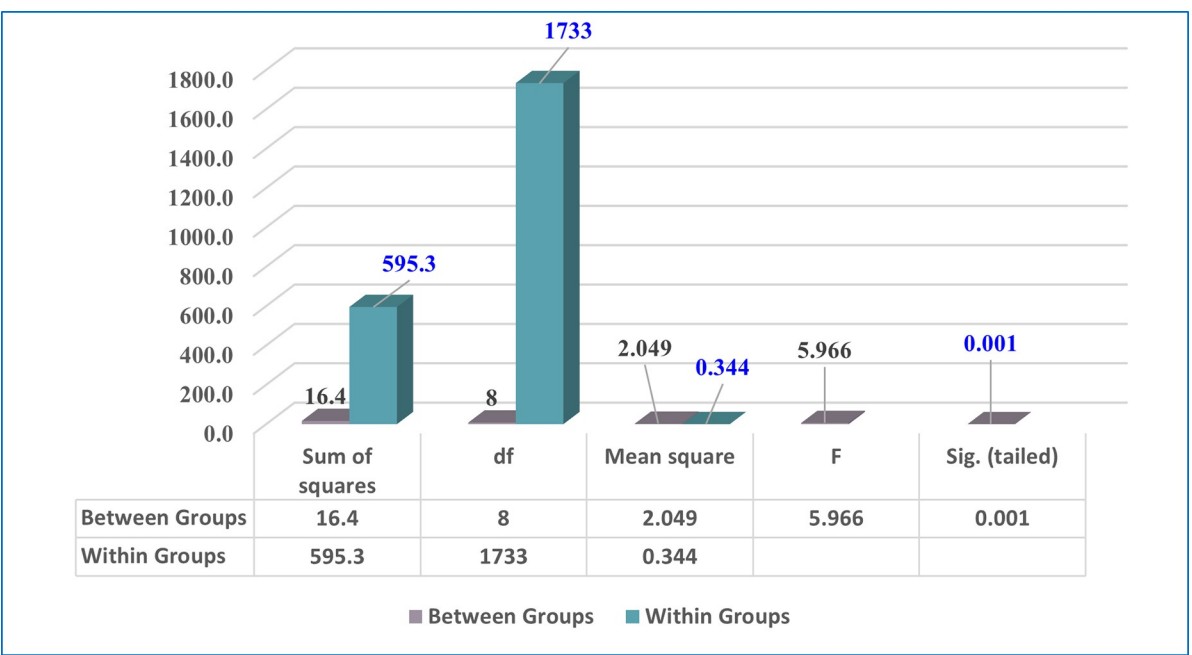

**Fig 6. One-way ANOVA test for college variable among students.**

being clear and easy, and the ability of learners to take the exam anywhere and at any time. These results are consistent with those of previous studies [1, 3, 5–8, 17, 24, 30–33, 44]. The results of these studies indicated that students showed their openness to and acceptance of e-exams, and they also confirmed that they prefer e-exams, especially when the type of questions

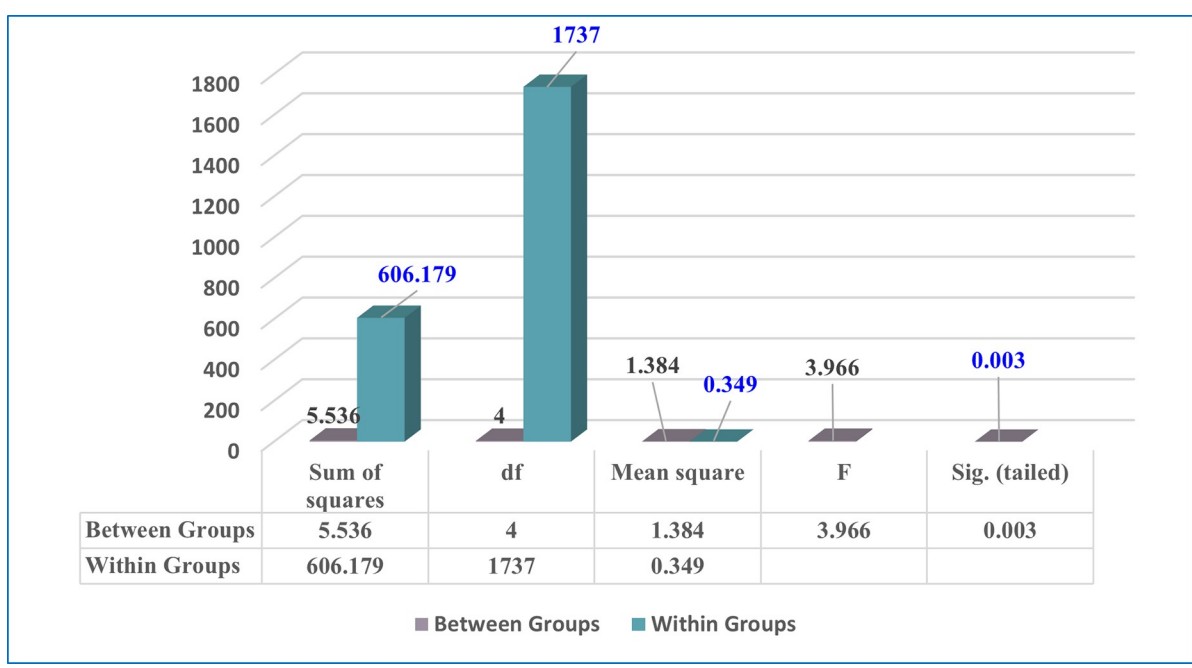

**Fig 7. One-way ANOVA test for academic year variable among students.**

**Table 8. The Scheffe test results according to the college variable.**

| (I) The college | (J) The college | Mean difference (I-J) | Sig. |
|---|---|---|---|
| Dentistry | Pharmacy & Health Science | .15351 | .722 |
| | Business Administration | -.01737 | 1.000 |
| | Engineering and Information Technology | -.15540 | .297 |
| | Architecture, Art and Design | -.05822 | .999 |
| | Law | -.23680* | .042 |
| | Mass and Communication | .03189 | 1.000 |
| | Humanities and Sciences | .00623 | 1.000 |
| | Medicine | -.03308 | 1.000 |
| Pharmacy &Health Science | Dentistry | -.15351 | .722 |
| | Business Administration | -.17089 | .571 |
| | Engineering and Information Technology | -.30892* | .004 |
| | Architecture, Art and Design | -.21174 | .408 |
| | Law | -.39031* | .000 |
| | Mass Communication | -.12162 | .951 |
| | Humanities and Sciences | -.14728 | .725 |
| | Medicine | -.18660 | .958 |
| Business Administration | Dentistry | .01737 | 1.000 |
| | Pharmacy & Health Science | .17089 | .571 |
| | Engineering and Information Technology | -.13803 | .461 |
| | Architecture, Art and Design | -.04085 | 1.000 |
| | Law | -.21943 | .082 |
| | Mass Communication | .04927 | 1.000 |
| | Humanities and Sciences | .02360 | 1.000 |
| | Medicine | -.01571 | 1.000 |
| Engineering and Information Technology | Dentistry | .15540 | .297 |
| | Pharmacy & Health Science | .30892* | .004 |
| | Business Administration | .13803 | .461 |
| | Architecture, Art and Design | .09718 | .952 |
| | Law | -.08139 | .981 |
| | Mass Communication | .18730 | .274 |
| | Humanities and Sciences | .16163 | .156 |
| | Medicine | .12232 | .996 |
| Architecture, Art and Design | Dentistry | .05822 | .999 |
| | Pharmacy & Health Science | .21174 | .408 |
| | Business Administration | .04085 | 1.000 |
| | Engineering and Information Technology | -.09718 | .952 |
| | Law | -.17858 | .525 |
| | Mass Communication | .09012 | .989 |
| | Humanities and Sciences | .06445 | .996 |
| | Medicine | .02514 | 1.000 |
| Law | Dentistry | .23680* | .042 |
| | Pharmacy & Health Science | .39031* | .000 |
| | Business Administration | .21943 | .082 |
| | Engineering and Information Technology | .08139 | .981 |
| | Architecture, Art and Design | .17858 | .525 |
| | Mass Communication | .26869* | .043 |
| | Humanities and Sciences | .24303* | .017 |
| | Medicine | .20371 | .913 |

*(Continued)*

**Table 8.** (Continued)

| (I) The college | (J) The college | Mean difference (I-J) | Sig. |
|---|---|---|---|
| Mass Communication | Dentistry | -.03189 | 1.000 |
| | Pharmacy & Health Science | .12162 | .951 |
| | Business Administration | -.04927 | 1.000 |
| | Engineering and Information Technology | -.18730 | .274 |
| | Architecture, Art and Design | -.09012 | .989 |
| | Law | -.26869* | .043 |
| | Humanities and Sciences | -.02566 | 1.000 |
| | Medicine | -.06498 | 1.000 |
| Humanities and Sciences | Dentistry | -.00623 | 1.000 |
| | Pharmacy & Health Science | .14728 | .725 |
| | Business Administration | -.02360 | 1.000 |
| | Engineering and Information Technology | -.16163 | .156 |
| | Architecture, Art and Design | -.06445 | .996 |
| | Law | -.24303* | .017 |
| | Mass Communication | .02566 | 1.000 |
| | Medicine | -.03932 | 1.000 |
| Medicine | Dentistry | .03308 | 1.000 |
| | Pharmacy & Health Science | .18660 | .958 |
| | Business Administration | .01571 | 1.000 |
| | Engineering and Information Technology | -.12232 | .996 |
| | Architecture, Art and Design | -.02514 | 1.000 |
| | Law | -.20371 | .913 |
| | Mass Communication | .06498 | 1.000 |
| | Humanities and Sciences | .03932 | 1.000 |

* Statistically significant at ($p < 0.05$)

are multiple-choice or true/false. Moreover, the results agree with studies that confirmed that students prefer e-exams because they provide marks and feedback more quickly, and help them to improve their learning and understanding of the content of the curriculum compared to traditional paper exams [6, 7, 27, 44]. In contrast, however, some of the students' responses to the questionnaire items showed negative attitudes towards the implementation and application of e-exams at their university during the Covid-19 pandemic spread. For example, they responded to Item -24 ('Internet interruption while I am doing e-exams causes me great anxiety') with a high level, with a mean value of 3.72. This means that undergraduate students at Ajman University might be feeling anxiety and stress as a result of carrying out e-exams rather than traditionally printed examination papers. This result is consistent with previous studies that found that students were upset, confused, and nervous about their inability to explain their

**Table 9. One-way ANOVA test for academic year variable among students.**

| | | Sum of squares | df | Mean square | F | Sig. (tailed) | Sig. level |
|---|---|---|---|---|---|---|---|
| Academic year | **Between Groups** | 5.536 | 4 | 1.384 | 3.966 | 0.003* | Significant |
| | **Within Groups** | 606.179 | 1737 | .349 | | | |
| | **Total** | 611.715 | 1741 | | | | |

* Statistically significant at ($p < 0.05$)

**Table 10. The results of the Scheffe test according to the academic year variable.**

| (I) Academic Year | Mean Difference (I-J) | Sig. | (I) Academic Year |
|---|---|---|---|
| First | Second | .03336 | .922 |
| | Third | -.08773 | .253 |
| | Fourth | .11649 | .289 |
| | Fifth | .31250 | .968 |
| Second | First | -.03336 | .922 |
| | Third | -.12109 | .077 |
| | Fourth | .08313 | .683 |
| | Fifth | .27914 | .979 |
| Third | First | .08773 | .253 |
| | Second | .12109 | .077 |
| | Fourth | .20422* | .012 |
| | Fifth | .40023 | .923 |
| Fourth | First | -.11649 | .289 |
| | Second | -.08313 | .683 |
| | Third | -.20422* | .012 |
| | Fifth | .19601 | .995 |
| Fifth | First | -.31250 | .968 |
| | Second | -.27914 | .979 |
| | Third | -.40023 | .923 |
| | Fourth | -.19601 | .995 |

* Statistically significant at ($p < 0.05$)

responses and answers during e-exams, due to strict computer technology settings [14, 37–39, 45]. Furthermore, Ajman University undergraduate students' responses to Item -26 ('I feel that it is easy to cheat while performing e-exams') also came with a high level, with a mean value of 3.70. This might mean that undergraduate students at Ajman University might feel that there may be opportunities for some cases to cheat while students perform e-exams. This result may be consistent with the study conducted by [40] Comas-Forgas et al. (2021), who pointed out that learners thought it easier to cheat when doing electronic examinations online. Also, it is consistent with the results obtained by [9] Da'asin (2016), who pointed out that e-exams might be a reliable and competent tool to measure what they aim to measure, but they may raise learners' anxiety and tension levels and might make cheating easier. Moreover, Ajman University undergraduate students' responses to Item-10 ('I think the electronic exam system was successful in protecting against technical problems') came with the lowest mean value of 2.54. This might mean that undergraduate students at Ajman University might have faced some technical problems while taking their e-exams during the spread COVID-19 pandemic through the second semester of the academic year 2019/2020. This result might be consistent with the studies conducted by [37], who pointed out that learners were upset and confused about not being able to answer e-exams questions due to strict computer technology settings.

The second research question focused on determining whether the degree of acceptance of e-exams by undergraduate students at Ajman University during the spread COVID-19 varied, from the students' perspectives, according to gender, college, and academic year. Our findings (as seen in Tables 6–10, and Figs 5–7) showed that the degree of acceptance of e-exams by undergraduate students varies according to gender in favor of females. This result means that female Ajman University undergraduate students' acceptance of e-exams during the spread COVID-19 is greater than the acceptance of their male counterparts. Also, the results indicate

that acceptance also varies according to college type (in favor of the Pharmacy & Science College), and according to academic year (in favor of the third academic year).

## 6. Conclusion

Due to the spread of the COVID-19 pandemic, most educational institutions, such as universities and schools, have moved towards using technology in the process of assessing students through the implementation of e-exams during the learning and teaching process. It might that a large-scale shift towards e-exams can be expected during the next few years if the COVID-19 pandemic is not completely controlled in the world. The current study aimed to explore the acceptance of e-exams on the part of Ajman University undergraduate students during the spread of the COVID-19 pandemic. Findings showed that Ajman University undergraduate students demonstrated a moderate acceptance of e-exams during the spread COVID-19 pandemic, with a total average and SD respectively of 3.18 and 1.22. This may indicate that some students accepted e-exams and they gave a positive impression about it in their responses to the items of the questionnaire. This impression might due to the positive advantages that are distinguished by them, such as quicker feedback and marks, time savings, environmentally friendly, easy to identify and access unanswered questions, the system of e-exams being clear and easy, and the ability of learners to take the exam anywhere and at any time. At the same time, responses of the undergraduate students to some other questionnaire items showed negative attitudes towards the implementation and application of e-exams at their university during the Covid-19 pandemic. This impression might due to their feelings of anxiety and stress as a result of completing e-exams rather than traditionally printed examination papers. Also, it may be due to the fact that they were confused and nervous about their inability to explain their responses and answers during the electronic exam, due to strict computer technology settings. Furthermore, the findings indicated that female Ajman University undergraduate students' acceptance of e-exams during the spread COVID-19 is greater than the acceptance of their male counterparts. Also, the results indicate that acceptance also varies according to college type (in favor of the Pharmacy & Health Science College), and according to academic year (in favor of the third academic year). Like any other analysis, this study has some limitations that should be acknowledged.

- This study was limited to students' responses, and the responses of faculty members were not taken. This is because the authors have investigated the level of acceptance of e-exams by undergraduate students at Ajman University during the spread of COVID-19, since the students consider are the center of the educational system and their perspectives and impressions are of high importance for improving the processes of assessment and evaluation.

- The study was limited to a sample size of 1986 students, representing 30% of the study population.

- The study conducted in main campus of Ajman University during and the tool of study (questionnaire) was distributed via email, WhatsApp, and other social Media during the second semester of the academic year 2019/2020.

## 7. Implications and recommendations

Notwithstanding the aforementioned limitations, the following suggested educational implications and recommendations are provided for future research on the implementation of e-exams during the spread of the COVID-19 pandemic.

- Most higher education institutions adopted the decision to temporarily avoid all in-person contact and close their campuses completely during the COVID-19 pandemic, which led

these institutions to apply online e-exams instead of traditional paper exams. Thus, it necessary to provide students with accurate and fair grades. This requires universities to provide a protection system for these e-exams on an ongoing basis.

- Appropriate solutions need to be found to technical problems and the disruption of the internet during the implementation of e-exams.

- It is necessary to provide technical support on an ongoing basis while conducting electronic tests.

- It is necessary that processes be established to ensure that there are no cases of cheating during electronic examinations.

- Similar research should be performed on the implementation of e-exams in higher education institutions.

## Supporting information

**S1 Data.**
(XLSX)

## Acknowledgments

The authors' would like to thank Ajman University for its cooperation, and the Dean of Scientific Research, for his guidance and mentorship.

## Author Contributions

**Formal analysis:** Najeh Rajeh Alsalhi.

**Investigation:** Sami Sulieman Al-Qatawneh.

**Methodology:** Najeh Rajeh Alsalhi.

**Project administration:** Sami Sulieman Al-Qatawneh.

**Resources:** Sami Sulieman Al-Qatawneh.

**Software:** Najeh Rajeh Alsalhi.

**Validation:** Mohd. Elmagzoub Eltahir.

**Writing – original draft:** Najeh Rajeh Alsalhi.

**Writing – review & editing:** Mohd. Elmagzoub Eltahir, Sami Sulieman Al-Qatawneh.

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
