## [Decision Letter · Decision Letter 0]

19 Nov 2021

PONE-D-21-20789Implementation of electronic exams during the spread of the COVID-19 pandemic: A quantitative study in higher educationPLOS ONE

Dear Dr. Alsalhi,

Thank you for submitting your manuscript to PLOS ONE. After careful consideration, we feel that it has merit but does not fully meet PLOS ONE’s publication criteria as it currently stands. Therefore, we invite you to submit a revised version of the manuscript that addresses the points raised during the review process.

Please see the reviewer comments for the minor revisions required before this manuscript can be published.==============================

We look forward to receiving your revised manuscript.

Kind regards,

Christine E. King, PhD

Academic Editor

PLOS ONE

2. Please provide additional details regarding participant consent. In the ethics statement in the Methods and online submission information, please ensure that you have specified whether consent was informed.

Reviewers' comments:

Reviewer's Responses to Questions

**Comments to the Author**

1. Is the manuscript technically sound, and do the data support the conclusions?

Reviewer #1: Partly

Reviewer #2: Yes

2. Has the statistical analysis been performed appropriately and rigorously? 

Reviewer #1: Yes

Reviewer #2: Yes

3. Have the authors made all data underlying the findings in their manuscript fully available?

Reviewer #1: Yes

Reviewer #2: Yes

4. Is the manuscript presented in an intelligible fashion and written in standard English?

Reviewer #1: Yes

Reviewer #2: No

5. Review Comments to the Author

Reviewer #1: 1- Modify some terms:

Electronic Exam e-exam

during the spread of the COVID-19 pandemic during the COVID-19 pandemic

2- The faculty's attitudes toward e-exams were discussed in former research. The authors should explain why faculty's attitudes of using e-exams considered to be explored again during the COVID-19 pandemic.

3- The researchers did not take the influence of criteria for which the e-exam were designed, as well as the capabilities of the faculty members in designing the e-exam (especially since they are from specializations that do not have an electronic reference).

4- Research questions, that drive the paper, should be built in the introduction from an ongoing and pertinent bibliography (up to 2021). These should be of global interest and not focused to a particular local problem. Identifying a research gap is not enough; key is showing its significance to the field. It is better to get recent studies during the Corona period, because there are a large number of studies that have been done and dealt with axes close to the topic of research.

5- Answer your research question in the conclusions; what did we learn compared with current, significant research (up to 2021). The authors should make explicit suggestions about how their study affects the design or use of E-exam systems. Is there something new about a particular theory, or is there evidence of theory advancement?

6- How general are your results? These have to be of interest to the whole community. Relate these with your limitations. Please review the concept of research limitations such as history effects, sampling, etc.

7- Finally, what is the originality of this study? The authors should clarify the originality compared with other studies.

8- Explain how this paper differs from the related ones published in the technical literature.

9- - All references before 2010 are better to be replaced by modern references, especially in the discussion part of the search results.

10- Literature Review is very tall, need to summarize, then to add topic about Education challenges and exams during the Corona period.

11- It is preferable that the data be processed after dividing it based on the students’ specializations into three groups (humanitarian track – engineering track – health track) and each group is processed separately, because the specialization and related tasks can have a direct and significant impact.

12- The resolution for Figure 1. is very low.

13- There is no need for a diagram (figure 3 and 4) because it does not add anything new (just repeating) all the data are in the table (3 and 2).

14- The researchers did not clarify which program was used in designing the e-exam, and whether the same program was used, or whether each college used what it sees.

15- The researchers did not clarify whether the questionnaire was electronic or paper, and how it was distributed (whether by e-mail or otherwise).

Reviewer #2: The author should contact language expert to update it as i have seen some mistakes in it. Moreover more references should be giving in literature review as well as in discussion section. Fresh references should be added in the whole article.

6. PLOS authors have the option to publish the peer review history of their article (what does this mean?). If published, this will include your full peer review and any attached files.

Reviewer #1: **Yes: **usama mohamed Abd Elsalam ibrahem

Reviewer #2: **Yes: **Dr. Qaisar Abbas

---

## [Author Response · Author response to Decision Letter 0]

10 Mar 2022

Authors response to reviewers and editor comments, they attached their response in the file named (Response to Reviewers), also, they attached the (Revised Manuscript with Track Changes) and (Revised Manuscript).

---

## [Editor Report · Decision Letter 1]

31 Mar 2022

Implementation of E-exams During the COVID-19 Pandemic: A quantitative Study in Higher education

PONE-D-21-20789R1

Dear Dr. Alsalhi,

We’re pleased to inform you that your manuscript has been judged scientifically suitable for publication and will be formally accepted for publication once it meets all outstanding technical requirements.

Kind regards,

Christine E. King, PhD

Academic Editor

PLOS ONE
---

## [Editor Report · Acceptance letter]

12 May 2022

PONE-D-21-20789R1 

Implementation of E-exams During the COVID-19 Pandemic: A quantitative Study in Higher education 

Dear Dr. Alsalhi:

I'm pleased to inform you that your manuscript has been deemed suitable for publication in PLOS ONE. Congratulations! Your manuscript is now with our production department. 

Kind regards, 

on behalf of

Dr. Christine E. King 

Academic Editor

PLOS ONE